psychology/neuroscience

adaptation, direction after-effect, motion processing, visual perception

**Author for correspondence:**
William Curran
e-mail: w.curran@qub.ac.uk

# The direction after-effect is a global motion phenomenon

William Curran, Lee Beattie, Delfina Bilello,
Laura A. Coulter, Jade A. Currie
and Jessica M. Pimentel Leon

School of Psychology, Queen's University Belfast, David Keir Building, 18–30 Malone Road, Belfast BT9 5BN, UK

 WC, 0000-0002-8821-8658

Prior experience influences visual perception. For example, extended viewing of a moving stimulus results in the misperception of a subsequent stimulus's motion direction—the direction after-effect (DAE). There has been an ongoing debate regarding the locus of the neural mechanisms underlying the DAE. We know the mechanisms are cortical, but there is uncertainty about where in the visual cortex they are located—at relatively early local motion processing stages, or at later global motion stages. We used a unikinetic plaid as an adapting stimulus, then measured the DAE experienced with a drifting random dot test stimulus. A unikinetic plaid comprises a static grating superimposed on a drifting grating of a different orientation. Observers cannot see the true motion direction of the moving component; instead they see pattern motion running parallel to the static component. The pattern motion of unikinetic plaids is encoded at the global processing level—specifically, in cortical areas MT and MST—and the local motion component is encoded earlier. We measured the direction after-effect as a function of the plaid's local and pattern motion directions. The DAE was induced by the plaid's pattern motion, but not by its component motion. This points to the neural mechanisms underlying the DAE being located at the global motion processing level, and no earlier than area MT.

## 1. Introduction

Motion encoding by the visual cortex is a hierarchical process, with local motion information extracted during the early stages of visual processing and global motion information extracted at the later stages [1–4]. This local–global division of labour by the visual cortex is a consequence of the increasing receptive field size of neurons as information travels along the motion-processing pathway [5], with receptive field size of motion-processing neurons in area V5/MT being about three times larger than that of motion-processing neurons in area V1 at any given eccentricity

[6]. As well as this cortical segregation of local and global motion processing it has been shown that the processing of complex motion patterns is reserved for later stages of the motion-processing pathway, with rotating and expanding/contracting motion patterns not being encoded until area MT+ [7–9]. While area MT is associated with global motion processing, there is some evidence that global motion is not fully represented at the level of individual MT neurons [10,11]. For example, Majaj et al. [10] report that individual MT neurons that responded selectively to a plaid pattern drifting within their receptive fields lost this selectivity when the plaid's component gratings were presented at different locations within each neuron's receptive fields. While the pattern motion of a drifting plaid is encoded at the individual MT neuron level, Majaj et al.'s results suggest this only applies for plaid stimuli whose components are present at the same spatial location.

Because of its hierarchical structure, a question often asked about motion-induced phenomena is where in the motion-processing hierarchy are the neural mechanisms underlying them to be found. For instance, functional magnetic resonance imaging studies [12–15] have identified cortical area V5/MT as a likely location for neural mechanisms underlying the motion after-effect (MAE)—a well-known phenomenon in which one perceives illusory movement following prolonged viewing of a moving pattern. However, psychophysics studies have revealed a more complex story of the neural mechanisms underlying the MAE. These studies suggest that the static MAE (which uses a static test stimulus following adaptation) reflects neural adaptation of low-level motion-sensitive neurons [16–18], which are found in cortical areas V1 and V2. The dynamic MAE (which uses dynamic test stimuli such as temporal flicker), on the other hand, is thought to reflect adaptation of higher-level motion-sensitive neurons [18,19] in area V5/MT.

A range of other motion phenomena have been investigated with a view to identifying where neural mechanisms underlying them are to be found in the motion processing hierarchy. One such phenomenon, which is the focus of this paper, is the direction after-effect (DAE). With this phenomenon, prior adaptation to a unidirectional moving pattern results in an exaggerated perceived direction difference between the adaptor stimulus and a subsequently viewed test stimulus moving in a different direction. The effect is direction tuned with its magnitude peaking at an adaptor-test direction difference of approximately $30°–40°$ [20–22], and it is considered to be a consequence of reduced sensitivity to redundant (unchanging) information and the freeing-up of resources for coding changes that may occur in the environment [23,24]. The DAE undergoes interocular transfer [25,26], demonstrating that the effect is mediated by binocular cell adaptation. Since binocular-driven neurons are found only in the cortex, the DAE is clearly an expression of cortical mechanisms. However, what is not clear is whether the mechanisms underlying the DAE occur early in the visual cortex, at the level of local motion processing, or at the later global motion processing stage.

Curran et al. [25] investigated the cortical locus of the DAE by measuring its speed tuning function, and then using a number of mixed-speed adaptors in an attempt to test DAE magnitude predictions made by local and global models. Their results suggest the DAE is driven by adaptation of local motion processing mechanisms and the authors conclude that these results, along with their finding that the DAE undergoes incomplete (approx. 70%) interocular transfer, point to cortical area V1 as a strong candidate site for the neural mechanisms underlying the effect. The relevance of incomplete interocular transfer (IOT) is important since partial interocular transfer of after-effects is considered to reflect early cortical adaptation [19]. However, others [26] have reported close to 100% IOT for the direction after-effect; this suggests that the effect is driven by adaptation of binocularly sensitive neurons beyond layer 4Cα of V1, perhaps in MT. Consistent with this position is Schrater and Simoncelli's [22] evidence that the DAE is a consequence of adaptation at the global motion processing level. They had observers adapt to a grating stimulus prior to reporting the motion direction of a bikinetic plaid. A bikinetic plaid is a stimulus in which two drifting gratings with different orientations are superimposed. The percept is typically of a plaid pattern moving coherently in one direction rather than of the two component motions sliding transparently across one another. The pattern (or global) motion of plaids is known to be processed by pattern selective neurons in area MT, and the component (or local) motions are processed as early as area V1 [27]. Schrater and Simoncelli configured the plaid stimulus such that the resulting DAE would be in one direction (closer to the adaptor direction) if it were driven by adaptation operating at a local motion processing stage and in the opposite direction (away from the adaptor direction) if it were the result of adaptation at the global motion processing stage. The resulting DAE was consistent with the global processing account.

Further evidence for the DAE being a global motion phenomenon is reported by Kohn & Movshon [28], who explored the effects of adaptation on neuronal direction tuning in macaque area MT. They found that adaptation in its preferred direction resulted in a narrowing of a neuron's tuning

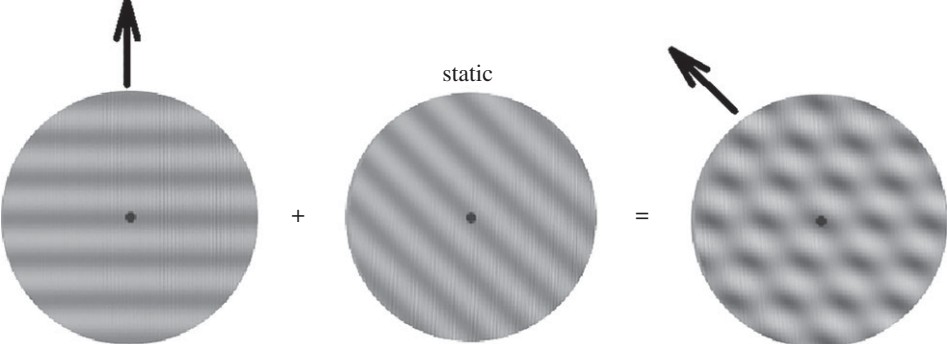

**Figure 1.** A unikinetic plaid is constructed by superimposing a moving grating (left) upon a static grating (centre). In this example the moving grating is drifting upwards and the static grating is oriented 45° from vertical. When superimposed to form a plaid the vertical component motion is invisible; instead, what one perceives is pattern motion running parallel to the static component (right). This particular configuration was used in the experiment along with three other configurations: a horizontal grating drifting upwards, and superimposed on a static grating oriented 22.5° or 67.5° from vertical; and a vertical static grating superimposed on a grating drifting 45° from vertically upwards (see text for details).

bandwidth, and adaptation on the flank of the neuron's tuning function resulted in direction tuning being shifted toward the adapted direction. Kohn and Movshon reason that, assuming direction-sensitive neurons retain their unadapted perceptual 'label', this attractive shift in tuning is consistent with the repulsive shift in perceived direction found with the DAE. They conclude that the DAE is driven by adaptation at the global motion processing level; however, they note that their results may also be modelled by weakening feedforward input from V1 into a recurrent model of MT circuitry.

As well as measuring the effects of prolonged presentation of a stimulus, the effects of other stimulus parameters on MT neuronal responsivity have been addressed. For example, motion-sensitive neurons in macaque MT rapidly increase their spiking in response to increasing dot density within their receptive fields, with responsivity plateauing off at a dot density of approximately 8 dots $deg^{-2}$ [29,30]. Similarly, macaque MT neurons' firing rates increase in a linear fashion in response to a stimulus's increasing motion coherence [31]. Motion coherence refers to the proportion of dots in a random dot stimulus that move in a single coherent direction. Motion stimuli that evoke a strong neural response also act as more effective adaptors [32] and, consequently, induce greater changes in neurons' direction tuning functions; the greater the change in tuning function the greater the magnitude of motion-adaptation effects [28,32], such as the DAE. Curran and Lynn [33,34] used this relationship between neuronal responsiveness and the strength of adaptation-induced motion distortions to explore the extent to which macaque MT responsivity mirrors that of its human homologue. They found that, as predicted by the macaque data, DAE magnitude increased with increasing adaptor dot density before plateauing off at approximately 10 dots $deg^{-2}$ [33]. In a similar vein, DAE magnitude increased linearly with increasing adaptor motion coherence signal [34]. Given that these results were predicted based on the known response characteristics of MT neurons in area MT, they support the position that the DAE is driven by adaptation at the global motion processing level.

While a consensus has yet to be made regarding the location of the neural mechanisms underlying the DAE, there is a larger body of evidence in support of the global processing position than the local processing position. Here we describe an experiment which attempts to settle the debate by using a unikinetic plaid as the adaptor stimulus. Like its bikinetic counterpart, a unikinetic plaid comprises two gratings that differ in their orientations; unlike the bikinetic plaid, one of the gratings in the unikinetic plaid remains stationary and the other one drifts at a constant speed. When positioned in central vision one does not perceive the true motion direction of the drifting grating, but instead perceives a pattern motion whose direction runs parallel to the orientation of the stationary grating (figure 1). In other words, the unikinetic plaid contains two motion directions—an invisible component motion direction and a visible pattern motion direction. It has been shown that cortical areas MT and MST (which receives input from MT) contain neurons that respond selectively to the pattern direction of unikinetic plaids, and that area MST contains a larger proportion of these pattern-sensitive neurons than area MT [35].

In our experiment observers adapted to a unikinetic plaid containing either a vertical stationary grating and a moving grating which drifted in a non-vertical direction, or a non-vertical stationary grating with

one of three orientations and a moving grating which drifted upwards. The former stimulus appeared to drift vertically upwards, and the latter appeared to drift in a non-vertical direction. Following adaptation, observers judged the direction of a random dot test pattern relative to vertically upwards. The occurrence of a DAE following adaptation to the plaid stimuli with non-vertical pattern motion would be clear evidence that the effect was induced by the adaptor's pattern motion rather than its component motion (which drifted vertically upwards), and would point to area MT or area MST as a strong candidate site for adaptation underlying the DAE. If, on the other hand, a DAE effect is only induced by the vertically drifting adaptor, this would point unequivocally to the effect being induced by the adaptor's component motion. This is because, while the plaid appears to drift vertically upwards, its component motion actually drifts in a non-vertical direction. However, in this scenario one could not conclude that the resulting DAE was driven by local motion processing mechanisms; this is because neurons responsive to the component motion of plaids exist in both area V1 and MT [27]. Of course it is feasible that the resulting DAE will not be a consequence of adaptation to just the pattern motion or just the component motion, but may result from adaptation to both motions. If that is the case, then one would predict that the DAE will occur for all of the pattern motion directions, including the vertical pattern direction. A DAE would be expected to occur in the latter case because the component motion direction is non-vertical.

# 2. Methods

The experiment comprised two conditions—a baseline condition in which a drifting grating of varying orientations/directions was used as the adaptor stimulus to measure the direction tuning of the DAE, and the unikinetic condition in which the adaptor was a unikinetic plaid with varying component and global motion directions.

## 2.1. Participants

Nine participants (the authors and three naive) took part in the experiment. All participants had normal or corrected-to-normal vision.

## 2.2. Apparatus and stimuli

Stimuli were presented within a centrally positioned circular aperture (6.3° diameter) on a Sony GDM-F500R monitor driven by a Cambridge Research Systems VSG 2/5 graphics board at a frame rate of 120 Hz. In the baseline condition the adaptor stimulus was a grating which drifted at a speed of $3° \, s^{-1}$. In the unikinetic condition a static grating and a moving grating were superimposed, and the moving grating's speed was set such that the plaid's perceived pattern speed was $3° \, s^{-1}$. The gratings in both conditions had a spatial frequency of 1 cycle $deg^{-1}$ and a Michelson contrast of 0.6. The test stimulus in both conditions was a random dot kinematogram comprising equal numbers of black and white dots, with a mean luminance of 24.33 cd $m^{-2}$, a dot density of 12.9 dots $deg^{-2}$, a dot diameter of 1.8 arcmin, and a drift speed of $3° \, s^{-1}$.

## 2.3. Procedure

In the baseline condition, DAE magnitude was measured following adaptation to a drifting sine grating with one of four drift directions—0°, 22.5°, 45° and 67.5° from vertically upwards. The adaptor stimulus was initially presented for 30 s, after which a drifting random dot test stimulus was presented for 200 ms and observers had to judge whether the test stimulus's motion direction was left or right of vertically upwards. Both adaptor and test stimuli had a central spot to help maintain fixation. Participants were given one second to respond; if a response was not made by then the adaptor and test stimulus sequence was repeated. After the first adaptor-test trial, all subsequent test stimuli were preceded by 5 s of 'top-up' adaptation to ensure the adaptive state was maintained. Test stimulus direction was chosen by an adaptive method-of-constants procedure (adaptive probit estimation), a method that dynamically updates the set of stimuli being presented depending on the observer's previous responses [36]. The directions were selected to optimize the estimation of the 'point of subjective equality' (PSE); that is, the direction the test stimulus was moving when it was perceived as moving vertically up, and DAE magnitude was calculated as the difference between these two directions. Each PSE was derived from a block of trials comprising 64 adaptor-test pairings, with the same adaptor

direction being used throughout a block. While the same adaptor direction was used throughout a given block, the order of adaptor directions tested was randomized across blocks.

Each observer generated four psychometric functions for each adaptor direction; in the case of those adaptors whose directions were non-vertical (22.5°, 45° and 67.5°) half the psychometric functions were generated following adaptation to motion clockwise to vertical and half were generated following adaptation to motion counter-clockwise to vertical. This balancing of clockwise and counter-clockwise adaptor directions controlled for any potential difference between subjective and objective measures of vertical. The order in which the clockwise and counter-clockwise directions were tested was randomized across blocks of trials; i.e. one randomly numbered block would test with the 22.5° clockwise adaptor and another randomly numbered block would test with the 22.5° counter-clockwise adaptor.

The unikinetic condition was identical to the baseline condition with the exception that the adaptor was a unikinetic plaid. DAE magnitude was measured as a function of the unikinetic plaid's pattern motion direction—0°, 22.5°, 45° and 67.5°. Note that, in the 0° case, the static component had a vertical orientation and the moving component drifted at 45° from vertical. For the remaining directions the static component had a non-vertical orientation (22.5°, 45° or 67.5°) and the drifting component always moved vertically. If the DAE is driven by the plaid's component motion, then it should be present following adaptation to the 0° stimulus but not to any of the non-zero stimuli. This is because the drift direction of the component motion (45° from vertical) in the 0° condition is optimal for producing a DAE. The component motion's drift direction in the remaining adaptors, however, was vertically upwards—an adaptor direction which does not result in a DAE in an upwardly drifting test stimulus. If the DAE is driven by the plaid's pattern motion, then there will be no measurable DAE with the 0° adaptor but it will be present with the other, non-vertical adaptors.

# 3. Results

The results from the baseline and unikinetic conditions are shown in figure 2a,b, respectively. Figure 2a plots DAE magnitude as a function of the direction of the adapting grating stimulus, and figure 2b plots DAE magnitude as a function of the pattern direction of the unikinetic plaid adaptor. Figure 2a shows a steady increase in DAE magnitude as a function of increasing adaptor-test direction difference. A repeated measures ANOVA reveals a significant effect of adaptor direction in the baseline condition ($F_{3,24} = 32.09$; $p < 0.001$). The t-test analyses reveal that, in the case of the non-vertical adaptors, the actual direction of the test stimulus was significantly different from vertical when it was perceived as moving upwards (22.5°: $t_8 = 10.291$; $p < 0.001$; 45°: $t_8 = 8.98$; $p < 0.001$; 67.5°: $t_8 = 7.93$; $p < 0.001$). In the 0° case (adaptor drifting upwards) there was no measurable DAE ($t_8 = 0.487$; $p = 0.64$). Interestingly, the DAE magnitude is relatively small; it peaks at approximately 6°, which is much lower than previous reports of 11°–15° [20–22]. A probable explanation for this difference in effect magnitude is that, unlike previous studies, we used different adaptor and test stimuli (gratings and dot patterns).

Figure 2b plots DAE magnitude for the unikinetic condition, with DAE magnitude being plotted as a function of the unikinetic plaid's pattern direction. Note that the 0° direction denotes a unikinetic plaid in which the stationary component is vertical and the moving component's drift direction is non-vertical (45°). The remaining directions denote a non-vertical static component and vertical moving component. Again DAE magnitude increases as a function of adaptor direction and, unlike the baseline condition, peaks at 45° before dropping off. A repeated measures ANOVA reveals a significant effect of adaptor direction ($F_{3,24} = 28.23$; $p < 0.001$). t-Test analyses reveal significant DAE effects for non-zero adaptor pattern directions (22.5°: $t_8 = 10.296$; $p < 0.001$; 45°: $t_8 = 8.29$; $p = <0.001$; 67.5°: $t_8 = 9.468$; $p < 0.001$). There was no significant effect in the case of the zero direction adaptor relative to the 0° baseline condition ($t_8 = 1.99$; $p = 0.082$). Furthermore, DAE magnitude for the 22.5° and 45° plaid stimuli was not significantly different from the baseline condition (22.5°: $t_8 = 0.757$, $p = 0.47$; 45°: $t_8 = 0.381$, $p = 0.713$).

Whereas the DAE effect in the plaid condition (figure 2b) peaks for an adaptor direction of 45° this is not the case for the baseline condition, which shows the DAE increasing up to the largest adaptor direction used (67.5°). The t-test analysis shows the DAE magnitude induced by the 67.5° adaptor in the baseline condition to be significantly stronger than in the plaid condition ($t_8 = 3.33$; $p = 0.01$). The direction tuning pattern in the plaid condition is consistent with previous reports [20–22]. We tested five of the original participants (two authors and three naive) with the additional adaptor direction of

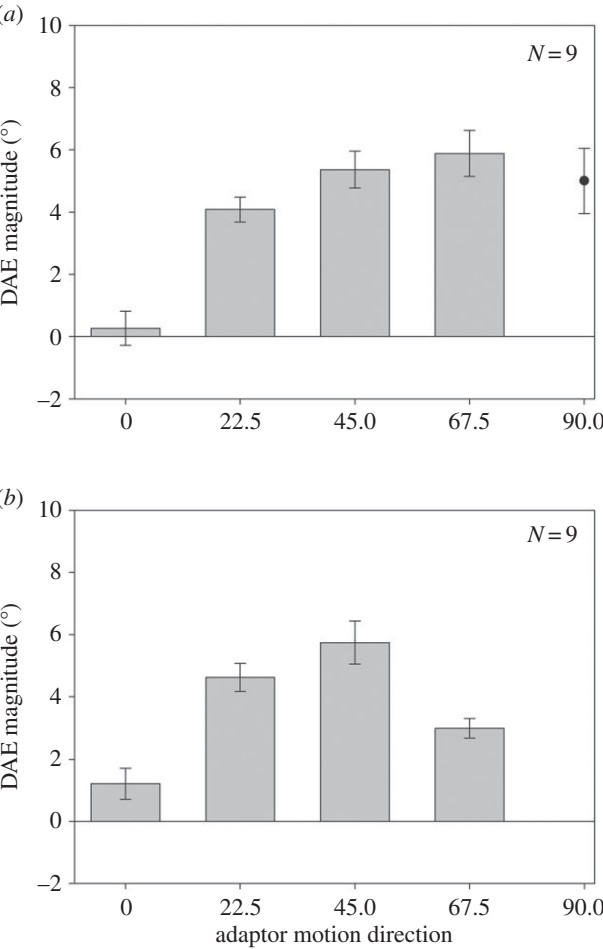

**Figure 2.** Direction after-effect magnitude as a function of adaptor motion direction in the (a) baseline condition and (b) unikinetic plaid condition (error bars denote $\pm 1$ s.e.). Note that the abscissa in (b) denotes the direction of the plaid's pattern motion. When the plaid pattern motion direction was 0° the component motion direction was 45° from vertical. For the remaining pattern motion directions, the component motion direction was 0° (vertically up).

90°, to check whether the baseline tuning function peaks at 67.5° or later. The results (filled circle, figure 2a) reveal that the baseline tuning function drops off after 67.5°.

The results depicted in figure 2b are consistent with the DAE being driven by adaptation of global motion mechanisms. When the adaptor's pattern motion direction was different from vertical (22.5°, 45° and 67.5°) and its component motion direction was vertical (0°), there was a significant DAE effect with a direction tuning that mirrored previous reports. When the pattern motion direction was upwards (0°), and its component motion direction was non-vertical (45°), the DAE effect was not significantly different from baseline, suggesting that local motion detector activity does not contribute to the DAE.

In the unikinetic plaid condition, component speed was set to ensure that the adaptor's pattern speed was the same ($3° \text{ s}^{-1}$) for all adaptor directions. Of course this means that the component speed varied across different adaptor directions, which is a potential confound. To address this issue we re-ran the unikinetic plaid condition of Experiment 1 with five participants (two authors and three naive) but this time kept the component drift speed fixed at $2.12° \text{ s}^{-1}$, which resulted in the pattern speed varying across adaptor directions. If the DAE is mainly driven by adaptation of global motion mechanisms, then we would expect results similar to the previous data. Figure 3 plots the data from this additional unikinetic plaid condition. The results are very similar to the original unikinetic plaid condition. They reveal that the DAE is tuned to the direction of the adaptor's pattern motion direction and the strength of the effect peaks at an adaptor direction of 45°. Repeated measures ANOVA reveals a significant effect of adaptor direction ($F_{3,12} = 21.12$; $p < 0.001$), and $t$-test analyses reveal a significant DAE in the non-vertical adaptor conditions only (0°: $t_4 = 0.996$; $p = 0.376$; 22.5°: $t_4 = 11.504$; $p < 0.001$; 45°: $t_4 = 9.599$; $p < 0.001$; 67.5°: $t_4 = 5.95$; $p = 0.004$).

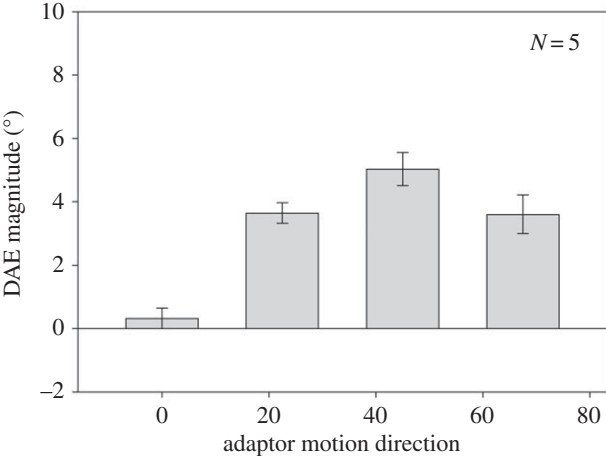

**Figure 3.** Direction after-effect magnitude as a function of the unikinetic plaid adaptor's pattern motion direction. In contrast to figure 2b, these data were generated following adaptation to a unikinetic plaid in which the component motion speed was kept constant across all directions tested.

## 4. Discussion

The direction after-effect is a well-known visual illusion in which prior exposure to unidirectional motion distorts the perceived direction of a subsequent test stimulus moving in a different direction, such that the test stimulus's apparent direction is repelled from the adapting direction. It is thought that the effect, and other adaptation-induced visual effects, is a consequence of the visual system reducing its sensitivity to redundant information which, in turn, frees up resources for coding changes that occur in the environment [23,24]. We know from previous research that the neural mechanisms that drive the DAE are cortical [25,26], but to date there has been no consensus on where in the cortex these mechanisms are to be found—at the level of local motion processing (e.g. area V1) or at the global motion processing level (MT+).

The aim of the current experiment was to settle the debate regarding the cortical location of the neural mechanisms underlying the DAE. To this end we used an adaptor, a unikinetic plaid, which we argue is ideal for separating the effects of local and global motion adaptation. As described in the Methods, a unikinetic plaid comprises a stationary grating superimposed on a differently oriented moving grating. When viewing a unikinetic plaid, one does not see the true motion direction of the moving component; rather, what is seen is a pattern motion whose direction is parallel to the stationary component. We tested whether the DAE induced by a unikinetic plaid is a result of adaptation to the plaid's component motion or its pattern motion. The results are compelling. When the pattern motion direction of the plaid was vertical and the component motion direction was 45° from vertical, there was no significant DAE, thus ruling out local motion adaptation as the catalyst for the DAE. When the pattern motion was non-vertical and the component motion was vertical, there was a demonstrable and significant DAE; this points squarely to the DAE being driven by global motion adaptation. Furthermore, the direction tuning of the effect mirrored that reported by previous studies [20–22]. Interestingly, the strength of the DAE, in both the baseline and unikinetic conditions, is substantially weaker than previously reported, peaking at approximately 6° as opposed to previous reports of 11°–15°. As pointed out in the results section, the weakened effect observed in the present study is likely to be a consequence of using a test stimulus (random dot pattern) that was very different to the adaptors (sinewave grating, unikinetic plaid). Previous studies used the same stimulus type as adaptor and test, thus ensuring that both stimuli were processed by the same neural mechanisms. It is conceivable that the weakened DAE in the present study is a consequence of the adaptor and test stimuli partially overlapping, rather than completely overlapping, in terms of the neural mechanisms encoding them.

To ensure that the adaptor's pattern speed remained constant for all directions tested in the unikinetic plaid condition, it was necessary to vary the adaptor's component motion speed. To rule out this manipulation of component motion speed as a potential confound, we repeated the unikinetic plaid condition but this time kept the plaid's component motion speed constant across all pattern directions tested. The results were very similar to the original unikinetic plaid condition, reinforcing the conclusion that the observed DAE was driven by the adaptor's pattern motion direction.

The discussion up to this point has taken the position of an either-or dichotomy, in which the DAE induced by a unikinetic plaid adaptor is driven either by the adaptor's local component motion or by its pattern motion. Of course an alternative explanation is that the effect is driven by a combination of both adaptor properties, such as an inhibitory interaction between the component motion and pattern motion directions. If this were the case, then we would expect DAE magnitude to differ in the baseline and unikinetic plaid conditions. However, as discussed in the Results, there was no significant difference between the baseline and plaid conditions for three of the four directions tested (0°, 22.5° and 45°); this suggests that the observed DAE was not a consequence of an interaction between the component and pattern motions, but was driven by the pattern motion alone.

Our results and conclusions mark a sharp contrast with previous research from the same laboratory [25] suggesting that the DAE is driven by adaptation of local motion mechanisms. In that study Curran *et al*. had participants adapt to mixed-speed adaptors, and used the known characteristics of the DAE's speed tuning function to compare the resulting DAE magnitude against that predicted by local and global models. While it is not clear why the two studies resulted in conflicting data, one potential explanation [37] is that the DAE measured in Curran *et al*.'s previous experiment was driven by the speed component of local motion signals rather than the local direction component. Using multi-element adaptors and test stimuli, Lee [37] went on to show that the propagation of local speed repulsion was indeed sufficient to produce a DAE.

While we used a unikinetic plaid adaptor in our experiment, our research question could have been addressed using a bikinetic plaid adaptor. For example, we could have used a bikinetic adaptor in which the two component motions drifted in orthogonal directions—0° and 90°. This would result in a pattern motion with a drift direction of 45°. If, as our results suggest, the DAE is an expression of adaptation at the global-motion processing level then adaptation to the bikinetic plaid should result in a DAE with a direction tuning function which peaks at approximately 45° from the pattern direction. If, on the other hand, the effect is driven by adaptation at the local-motion processing level then the resulting DAE's direction tuning function should peak 45° from either of the component motions (i.e. 90° from the pattern direction).

In summary, we used a unikinetic plaid to induce the direction after-effect in observers. This type of adaptor stimulus is ideal for separating the role of local and global motion processing mechanisms in the direction after-effect. Our results clearly demonstrate that the direction after-effect is driven by the pattern motion, but not the component motion, of unikinetic plaids. Given what is known about where in the cortex the pattern motion of unikinetic plaids is encoded [35], we conclude that the direction after-effect is driven by adaptation of neural mechanisms in either area MT or MST (or both)—cortical areas that are considered to be involved in global motion processing.

Ethics. The experiments were approved by the School of Psychology Ethics Committee at Queen's University Belfast. The experiment was conducted in accordance with the Code of Ethics of the World Medical Association (Declaration of Helsinki) and informed consent was obtained from all participants.

Data accessibility. Data from the experiments are available from the Dryad Digital Repository at: https://doi.org/10.5061/dryad.217cc62 [38].

Authors' contributions. All authors contributed to experimental design and running the experiments. L.B. analysed the data. W.C. drafted the manuscript.

Competing interests. We declare we have no competing interests.

Funding. This research was not supported by external funding.

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
