## [Reviewer comments · Royal Society Open Science]

Review History

RSOS-181271.R0 (Original submission)

Review form: Reviewer 1

Is the manuscript scientifically sound in its present form?

Yes

Are the interpretations and conclusions justified by the results?

Yes

Is the language acceptable?

Yes

Is it clear how to access all supporting data?

Yes

Do you have any ethical concerns with this paper?

No

Have you any concerns about statistical analyses in this paper?

No

Recommendation?

Accept as is

Comments to the Author(s)

The manuscript investigates the level in the visual system that the direction aftereffect (DAE) occurs. The two levels investigated are the local (V1) and the global (V5) motion levels. These two levels are teased apart by comparing the DAE tuning functions produced by a unikinetic plaid and a sinewave grating. The perceived direction of a unikinetic plaid is in the direction of the static grating and its perception has been linked to V5 and above. The logic of the study was that if the DAE is driven by the local-motion cells, then the tuning function for the unikinetic plaid should be relative to the local-motion direction (the direction of the moving grating) whereas if it is driven by global-motion cells it should be relative to the plaid direction (parallel to the static grating). Results clearly indicate that the plaid direction drives the DAE, and hence supports a global-motion locus to the DAE.

This is an elegant study that nicely addresses the question of what level in the motion system drives the DAE. The manuscript is well written and clear and I have no concerns with it.

Review form: Reviewer 2 (Alan Lee)

Is the manuscript scientifically sound in its present form?

No

Are the interpretations and conclusions justified by the results?

No

Is the language acceptable?

Yes

Is it clear how to access all supporting data?

Yes

Do you have any ethical concerns with this paper?

No

Have you any concerns about statistical analyses in this paper?

No

Recommendation?

Major revision is needed (please make suggestions in comments)

Comments to the Author(s)

The authors used drifting gratings and unikinetic plaids to assess whether DAE comes from the local or global level of motion processing, and their findings support the global-level side. The experiment was carefully conducted, but there seems to be a possible confound that needs to be handled before the authors could draw such conclusion. Along this line, the authors may want to present more data in order to draw a more convincing conclusion (currently, there are only data

from one experiment...). Also, the writing (especially in terms of connecting the present study with the literature) and some analyses needs to be improved. Below I list the major and minor issues to be addressed by the authors.

Major issues:

1. drifting speed of the component as a confound: The authors were careful in controlling the component drifting speed in the unikinetic condition to produce the same perceived pattern speed across different adaptor directions. But that means the component drifting speed varied across adaptor directions, which was an obvious confound. One way to address the issue is to rerun the unikinetic condition with constant component drifting speed so that the pattern speed varied across adaptor directions. If the results were similar with the original unikinetic condition, then the main driving factor should be the "perceived pattern direction of the adaptor". But if the results were different, it may require further investigation.

2. a loosely-connected Discussion: The author may consider focusing the Discussion on discussing the present study and the results, especially the limitations of the experiments. Currently, some of the Discussion (especially the first and second paragraphs) sounds like repeating some parts of the Introduction in reviewing previous findings, with no connection to the present study.

For example, the authors may want to suggest how readers can reconcile the apparently contradictory conclusions drawn in the present study and that in Curran et al. (2006). Also, the definition for "global" and "local" may need further clarification, as, currently, "global" only refers to the "pattern direction from a plaid", without addressing other "global" characteristics such as cross-location integration, large receptive field size, etc. Lastly, the authors may want to include two relevant studies from Tony Movshons lab to enrich the discussion/intro (Majaj, Carandini, and Movshon, 2007, which shows that motion integration in MT could be location-specific; and Tailby, Majaj, and Movshon, 2010, which may be related to the authors' discussion on interocular transfer of DAE).

3. inadequate analysis and interpretation on the results: First, there seems to be a significant difference in DAE in between baseline and unikinetic at 67.5 degrees (i.e., the rightmost bar in each of the two figures). The author should report the test of difference and provide potential explanation if the difference is significant (which, I suppose, is the case, by eyeballing the graphs). Second, if this difference is indeed significant, they authors may want to comment on the overall trend of the DAE, especially in the baseline condition, because it was not consistent with those in the literature, in which DAE typically peaks at 45 degrees (e.g., Schrater and Simoncelli, 1998).

4. How about bikinetic plaids? If DAE is driven by the "global-motion" detectors, a bikinetic plaid with, say, upward and leftward drifting components with identical speed should signal a 45-degree top-left global motion direction. If an observer adapts to this bikinetic pattern, there should be significant DAE on a test stimulus presented at the upward (0-degree) direction. In fact, I have a paper recently accepted by the Journal of Vision, addressing a very similar question asked by the authors in the present study, but I used a different approach: a multiple-aperture stimulus (Amano et al., 2009, JoV) with multiple global directions embedded in the adaptor (Lee and Lu, 2014, AP&P). In one of the experiments, I used bikinetic plaid as both the adaptor and test stimulus, but I did not find a stronger DAE when compared to a single-grating case (for both adaptor and test). Although the general conclusion of my paper is in line with that of the present study (that DAE relies on global-level adaptation, especially when tested with a global-level stimulus), I believe this bikinetic plaid condition would be interesting to explore, and should be discussed by the authors, given that they used plaid as their main stimulus.

Minor issues

- Page 10: for the unikinetic condition, why was the moving component drifted at 45 degrees only

in the 0-degree case? If I understand this correctly, a unikinetic plaid with a vertical static orientation and an upward drifting component should still be perceived as moving upward. The author would need to justify why they used a different tilt for the static orientation for the 0-degree, unikinetic case, or rerun this case with the above-described setup, so that the 0-degree case in the unikinetic condition could be more fairly compared with the other adaptor directions.

- What was the speed of the test RDK? This is an important piece of information because DAE is speed-tuned, and this is somewhat related to the Major issue number 1 above.

- Some details about the procedure need to be added, e.g., exact number of trials in each block, number of blocks completed by each observer, and the order of presentation of the adaptor directions.

- There were both clockwise and counter-clockwise (relative to vertical) directions for the non-vertical adaptors, but it was not described how these adaptor directions were presented. I suppose they were alternated or at least randomized across different blocks, but this needs to be clarified.

- I assume the authors took the estimated PSE as DAE magnitude. But this was not explicitly mentioned in method or results. Please add it back.

- The doi link at the bottom of the manuscript does not work for me.

Review form: Reviewer 3 (George Mather)

Is the manuscript scientifically sound in its present form?

Yes

Are the interpretations and conclusions justified by the results?

No

Is the language acceptable?

Yes

Is it clear how to access all supporting data?

Yes

Do you have any ethical concerns with this paper?

No

Have you any concerns about statistical analyses in this paper?

Yes

Recommendation?

Major revision is needed (please make suggestions in comments)

Comments to the Author(s)

Please see attached file (Appendix A).

Decision letter (RSOS-181271.R0)

27-Sep-2018

Dear Dr Curran:

Manuscript ID RSOS-181271 entitled "The direction aftereffect is a global motion phenomenon" which you submitted to Royal Society Open Science, has been reviewed. The comments from reviewers are included at the bottom of this letter.

In view of the criticisms of the reviewers, the manuscript has been rejected in its current form. However, a new manuscript may be submitted which takes into consideration these comments. This new manuscript should include additional data as requested by reviewers 2 and 3.

Please note that resubmitting your manuscript does not guarantee eventual acceptance, and that your resubmission will be subject to peer review before a decision is made.

Your resubmitted manuscript should be submitted by 27-Mar-2019. If you are unable to submit by this date please contact the Editorial Office.

Please note that Royal Society Open Science will introduce article processing charges for all new submissions received from 1 January 2018. Charges will also apply to papers transferred to Royal Society Open Science from other Royal Society Publishing journals, as well as papers submitted as part of our collaboration with the Royal Society of Chemistry (<http://rsos.royalsocietypublishing.org/chemistry>). If your manuscript is submitted and accepted for publication after 1 Jan 2018, you will be asked to pay the article processing charge, unless you request a waiver and this is approved by Royal Society Publishing. You can find out more about the charges at <http://rsos.royalsocietypublishing.org/page/charges>. Should you have any queries, please contact openscience@royalsociety.org.

on behalf of Dr Isabelle Mareschal (Associate Editor) and Prof. Antonia Hamilton (Subject Editor)
openscience@royalsociety.org

Associate Editor Comments to Author (Dr Isabelle Mareschal):

Associate Editor: 1

Comments to the Author:

Expert reviewers have read your manuscript and raised some important points that will require more data collection. For example reviewer 3 highlights the need for testing naive subjects and reviewer 2 suggests rerunning the unikinetic experiment. Both have suggestions for improving the readability/accessibility of the manuscript. Please provide a point by point reply to the reviewers.

Reviewers' Comments to Author:

Reviewer: 1

Comments to the Author(s)

The manuscript investigates the level in the visual system that the direction aftereffect (DAE) occurs. The two levels investigated are the local (V1) and the global (V5) motion levels. These two levels are teased apart by comparing the DAE tuning functions produced by a unikinetic plaid and a sinewave grating. The perceived direction of a unikinetic plaid is in the direction of the static grating and its perception has been linked to V5 and above. The logic of the study was that if the DAE is driven by the local-motion cells, then the tuning function for the unikinetic plaid should be relative to the local-motion direction (the direction of the moving grating) whereas if it is driven by global-motion cells it should be relative to the plaid direction (parallel to the static grating). Results clearly indicate that the plaid direction drives the DAE, and hence supports a global-motion locus to the DAE.

This is an elegant study that nicely addresses the question of what level in the motion system drives the DAE. The manuscript is well written and clear and I have no concerns with it.

Reviewer: 2

Comments to the Author(s)

The authors used drifting gratings and unikinetic plaids to assess whether DAE comes from the local or global level of motion processing, and their findings support the global-level side. The experiment was carefully conducted, but there seems to be a possible confound that needs to be handled before the authors could draw such conclusion. Along this line, the authors may want to present more data in order to draw a more convincing conclusion (currently, there are only data from one experiment...). Also, the writing (especially in terms of connecting the present study with the literature) and some analyses needs to be improved. Below I list the major and minor issues to be addressed by the authors.

Major issues:

1. drifting speed of the component as a confound: The authors were careful in controlling the component drifting speed in the unikinetic condition to produce the same perceived pattern speed across different adaptor directions. But that means the component drifting speed varied across adaptor directions, which was an obvious confound. One way to address the issue is to rerun the unikinetic condition with constant component drifting speed so that the pattern speed varied across adaptor directions. If the results were similar with the original unikinetic condition, then the main driving factor should be the "perceived pattern direction of the adaptor". But if the results were different, it may require further investigation.

2. a loosely-connected Discussion: The author may consider focusing the Discussion on discussing the present study and the results, especially the limitations of the experiments.

Currently, some of the Discussion (especially the first and second paragraphs) sounds like repeating some parts of the Introduction in reviewing previous findings, with no connection to the present study.

For example, the authors may want to suggest how readers can reconcile the apparently contradictory conclusions drawn in the present study and that in Curran et al. (2006). Also, the definition for “global” and “local” may need further clarification, as, currently, “global” only refers to the “pattern direction from a plaid”, without addressing other “global” characteristics such as cross-location integration, large receptive field size, etc. Lastly, the authors may want to include two relevant studies from Tony Movshons lab to enrich the discussion/intro (Majaj, Carandini, and Movshon, 2007, which shows that motion integration in MT could be location-specific; and Tailby, Majaj, and Movshon, 2010, which may be related to the authors’ discussion on interocular transfer of DAE).

3. inadequate analysis and interpretation on the results: First, there seems to be a significant difference in DAE in between baseline and unikinetic at 67.5 degrees (i.e., the rightmost bar in each of the two figures). The author should report the test of difference and provide potential explanation if the difference is significant (which, I suppose, is the case, by eyeballing the graphs). Second, if this difference is indeed significant, they authors may want to comment on the overall trend of the DAE, especially in the baseline condition, because it was not consistent with those in the literature, in which DAE typically peaks at 45 degrees (e.g., Schrater and Simoncelli, 1998).

4. How about bikinetic plaids? If DAE is driven by the “global-motion” detectors, a bikinetic plaid with, say, upward and leftward drifting components with identical speed should signal a 45-degree top-left global motion direction. If an observer adapts to this bikinetic pattern, there should be significant DAE on a test stimulus presented at the upward (0-degree) direction. In fact, I have a paper recently accepted by the Journal of Vision, addressing a very similar question asked by the authors in the present study, but I used a different approach: a multiple-aperture stimulus (Amano et al., 2009, JoV) with multiple global directions embedded in the adaptor (Lee and Lu, 2014, AP&P). In one of the experiments, I used bikinetic plaid as both the adaptor and test stimulus, but I did not find a stronger DAE when compared to a single-grating case (for both adaptor and test). Although the general conclusion of my paper is in line with that of the present study (that DAE relies on global-level adaptation, especially when tested with a global-level stimulus), I believe this bikinetic plaid condition would be interesting to explore, and should be discussed by the authors, given that they used plaid as their main stimulus.

Minor issues

- Page 10: for the unikinetic condition, why was the moving component drifted at 45 degrees only in the 0-degree case? If I understand this correctly, a unikinetic plaid with a vertical static orientation and an upward drifting component should still be perceived as moving upward. The author would need to justify why they used a different tilt for the static orientation for the 0-degree, unikinetic case, or rerun this case with the above-described setup, so that the 0-degree case in the unikinetic condition could be more fairly compared with the other adaptor directions.
- What was the speed of the test RDK? This is an important piece of information because DAE is speed-tuned, and this is somewhat related to the Major issue number 1 above.
- Some details about the procedure need to be added, e.g., exact number of trials in each block, number of blocks completed by each observer, and the order of presentation of the adaptor directions.
- There were both clockwise and counter-clockwise (relative to vertical) directions for the non-vertical adaptors, but it was not described how these adaptor directions were presented. I suppose they were alternated or at least randomized across different blocks, but this needs to be clarified.

- I assume the authors took the estimated PSE as DAE magnitude. But this was not explicitly mentioned in method or results. Please add it back.
- The doi link at the bottom of the manuscript does not work for me.

Reviewer: 3

Comments to the Author(s)
Please see attached file

Author's Response to Decision Letter for (RSOS-181271.R0)

See Appendix B.

RSOS-190114.R1 (Revision)

Review form: Reviewer 1

Is the manuscript scientifically sound in its present form?

Yes

Are the interpretations and conclusions justified by the results?

Yes

Is the language acceptable?

Yes

Is it clear how to access all supporting data?

Not Applicable

Do you have any ethical concerns with this paper?

No

Have you any concerns about statistical analyses in this paper?

No

Recommendation?

Accept as is

Comments to the Author(s)

I think the topic addressed by the manuscript is an interesting one, the experiments were well designed and conducted and the results make a worthwhile contribution to the field.

Review form: Reviewer 2 (Alan Lee)

Is the manuscript scientifically sound in its present form?

Yes

Are the interpretations and conclusions justified by the results?

Yes

Is the language acceptable?

Yes

Is it clear how to access all supporting data?

Yes

Do you have any ethical concerns with this paper?

No

Have you any concerns about statistical analyses in this paper?

No

Recommendation?

Accept as is

Comments to the Author(s)

The manuscript looks much better and the issues I raised have been adequately addressed.

Just one minor but general point about the tone: the authors seem quite confident in various parts in the manuscript that their results have definitively settled the debate (e.g., in Discussion, second paragraph, "this points squarely to the DAE being driven by global motion adaptation"). But apparently there are still some loose ends, e.g., as the authors acknowledged in discussing Curran et al.'s (2006) results. However, this may be more of an issue about writing style, and I don't have a very strong view on it. I'll leave this to the editor.

Decision letter (RSOS-190114.R1)

20-Feb-2019

Dear Dr Curran,

I am pleased to inform you that your manuscript entitled "The direction aftereffect is a global motion phenomenon" is now accepted for publication in Royal Society Open Science.

Royal Society Open Science operates under a continuous publication model (<http://bit.ly/cpFAQ>). Your article will be published straight into the next open issue and this

will be the final version of the paper. As such, it can be cited immediately by other researchers. As the issue version of your paper will be the only version to be published I would advise you to check your proofs thoroughly as changes cannot be made once the paper is published.

You have the opportunity to archive your accepted, unbranded manuscript, but access to the full text must be embargoed until publication.

Articles are normally press released. For this to be effective we set an embargo on news coverage corresponding to the publication date of the article. We request that news media and the authors do not publish stories ahead of this embargo (when final version of the article is available).

on behalf of Dr Isabelle Mareschal (Associate Editor) and Antonia Hamilton (Subject Editor)
openscience@royalsociety.org

Reviewer comments to Author:
Reviewer: 2

Comments to the Author(s)
The manuscript looks much better and the issues I raised have been adequately addressed.

Just one minor but general point about the tone: the authors seem quite confident in various parts in the manuscript that their results have definitively settled the debate (e.g., in Discussion, second paragraph, "this points squarely to the DAE being driven by global motion adaptation"). But apparently there are still some loose ends, e.g., as the authors acknowledged in discussing Curran et al.'s (2006) results. However, this may be more of an issue about writing style, and I don't have a very strong view on it. I'll leave this to the editor.

Reviewer: 1

Comments to the Author(s)
I think the topic addressed by the manuscript is an interesting one, the experiments were well designed and conducted and the results make a worthwhile contribution to the field.

Appendix A

I have some concern that all the participants are authors, so it is possible in principle for bias effects to contribute to the results. The psychophysical method was not bias-free because the response was explicitly about the perceived direction of the test pattern. It would be preferable to have data for naïve participants as well.

The predictions are presented in a way that is difficult to understand. Plaid adaptation stimuli were made by adding a static grating and a drifting grating at different orientations. The apparent direction of such a plaid is always parallel to the orientation of the static grating. In the four conditions employed (which I refer to by number 1-4), the apparent direction of the plaid should therefore be 0, 22, 45, and 67.5 deg with respect to vertical. The actual direction of the drifting grating in these conditions was 45, 0, 0, 0 deg respectively. Participants judged the apparent direction of a drifting random dot test pattern relative to vertical. Thus if the DAE is driven by plaid direction, it should be present in conditions 2, 3 and 4 but not in condition 1. An outcome in favour of this prediction would point towards involvement of MT or MST, according to the manuscript. If the DAE is driven by grating direction, it should be present in condition 1 but not in conditions 2, 3, and 4. This outcome would favour either V1 or MT. A baseline condition was also performed, in which the stimulus contains a single drifting grating drifting at 0, 22, 45 or 67.5 deg.

Results clearly show a major contribution to adaptation from the plaid direction. However, the claim that there is no effect of the local/grating direction is not so convincing. This claim requires that results for the zero-degree adapters in the baseline and plaid conditions should be the same, at zero. On the other hand, the presence of a DAE at a plaid direction of 0 deg would indicate some degree of adaptation to the 45 deg grating direction. The SE bars suggest an above-zero effect in this plaid condition (Fig. 2b). The results report a t-test that there was no significant effect in this condition. It is not clear what this t-test tested. Given that the same participants took part in all sessions, it would be possible to run a t-test to compare the zero deg adapters in the baseline and plaid conditions. Is this the t value reported? Perhaps more (naïve) participants would push this difference into significance?

If adaptation involves multiple components at different levels, one might expect some effect from the grating component. Indeed it could be argued

that there is so much evidence for adaptation at multiple levels of adaptation, including V1 and MT, that it is too simplistic to present the predictions in terms of an either-or dichotomy. There is evidence for adaptation in V1, so why would this not contribute to the DAE at all?

It could be argued in reply that the grating by itself at 45 deg produced a much larger effect (45 deg in the baseline condition, Fig 2a) than when it was present in the plaid (0 deg in the plaid condition, Fig 2b), but this could reflect the relative strengths of the two adapting components, as well as some inhibitory interaction between the static component and the drifting component. The point is at least worth debating.

Have the authors considered the possibility of an adaptation component mediated by second-order motion sensors that encode pattern direction in plaids? A number of papers, and a theoretical model, favour a role for second-order sensors in plaid processing and adaptation effects (e.g. Wilson, H. R., & Kim, J. 1994. A model for motion coherence and transparency. *Visual Neuroscience*, 11(6), 1205-1220; papers by Nishida cited in the manuscript).

Appendix B

Response to Reviewers (reviewers' comments italicized)

Reviewer #1

Reviewer #1 did not request any additional details and had no concerns.

Reviewer #2

Reviewer # 2 raised the following major issues.

1. *Drifting speed of the component as a confound: The authors were careful in controlling the component drifting speed in the unikinetic condition to produce the same perceived pattern speed across different adaptor directions. But that means the component drifting speed varied across adaptor directions, which was an obvious confound. One way to address the issue is to rerun the unikinetic condition with constant component drifting speed so that the pattern speed varied across adaptor directions. If the results were similar with the original unikinetic condition, then the main driving factor should be the “perceived pattern direction of the adaptor”. But if the results were different, it may require further investigation.*

Response. We have taken Reviewer #2's advice and re-run the unikinetic plaid condition while keeping component motion speed constant. The results are very similar to the original unikinetic plaid experiment. Details of this additional condition are included on pages 13 and 14.

2. *A loosely-connected Discussion: The author may consider focusing the Discussion on discussing the present study and the results, especially the limitations of the experiments. Currently, some of the Discussion (especially the first and second paragraphs) sounds like repeating some parts of the Introduction in reviewing previous findings, with no connection to the present study. For example, the authors may want to suggest how readers can reconcile the apparently contradictory conclusions drawn in the present study and that in Curran et al. (2006). Also, the definition for “global” and “local” may need further clarification, as, currently, “global” only refers to the “pattern direction from a plaid”, without addressing other “global” characteristics such as cross-location integration, large receptive field size, etc. Lastly, the authors may want to include two relevant studies from Tony Movshon's lab to enrich the discussion/intro (Majaj, Carandini, and Movshon, 2007, which shows that motion integration in MT could be location-specific;*

and Tailby, Majaj, and Movshon, 2010, which may be related to the authors' discussion on interocular transfer of DAE).

Response. We have turned our focus, as suggested, to the limitation of the experiment – see also response to point 1, above, and response to Reviewer #3's point 1. We have attempted to reconcile the apparent conflict between the current results and those of Curran et al. (2006), by appealing to Lee's (2008) point regarding local speed rather than local direction driving the DAE reported by Curran et al. (2006). This is addressed on pages 16 and 17. Reviewer #2 asked us for further clarification on the definitions for "global" and "local". In the Introduction we refer to the relationship between receptive field size and local/global distinction in motion processing, and we clearly emphasise that area MT+ is considered to be involved in global motion processing. The key point of our experiment is examining whether or not the DAE is determined by the pattern motion direction of unikinetic plaid adaptors, and thus driven by adaptation of motion-sensitive mechanisms in area MT+. Reviewer #2 also suggests we may wish to include two studies from Movshon's lab. These studies are certainly interesting, and we now refer to them in the first paragraph of the Introduction.

3. *Inadequate analysis and interpretation on the results: First, there seems to be a significant difference in DAE in between baseline and unikinetic at 67.5 degrees (i.e., the rightmost bar in each of the two figures). The author should report the test of difference and provide potential explanation if the difference is significant (which, I suppose, is the case, by eyeballing the graphs). Second, if this difference is indeed significant, they authors may want to comment on the overall trend of the DAE, especially in the baseline condition, because it was not consistent with those in the literature, in which DAE typically peaks at 45 degrees (e.g., Schrater and Simoncelli, 1998).*

Response. Reviewer #2 is correct in pointing out that there is a significant difference between baseline and unikinetic at 67.5 degrees, which is now reported. While we have not speculated on why the baseline condition does not peak at the same point as the unikinetic condition, we have established (through additional testing) that it does indeed peak and, consistent with previous reports, the observed DAE tuning function describes an inverted-U shape. This is addressed in the second paragraph of page 13.

4. *How about bikinetic plaids? If DAE is driven by the "global-motion" detectors, a bikinetic plaid with, say, upward and leftward drifting components with identical speed should signal a 45-degree top-left global*

motion direction. If an observer adapts to this bikinetic pattern, there should be significant DAE on a test stimulus presented at the upward (0-degree) direction. In fact, I have a paper recently accepted by the Journal of Vision, addressing a very similar question asked by the authors in the present study, but I used a different approach: a multiple-aperture stimulus (Amano et al., 2009, JoV) with multiple global directions embedded in the adaptor (Lee and Lu, 2014, AP&P). In one of the experiments, I used bikinetic plaid as both the adaptor and test stimulus, but I did not find a stronger DAE when compared to a single-grating case (for both adaptor and test). Although the general conclusion of my paper is in line with that of the present study (that DAE relies on global-level adaptation, especially when tested with a global-level stimulus), I believe this bikinetic plaid condition would be interesting to explore, and should be discussed by the authors, given that they used plaid as their main stimulus.

Response. As recommended by Reviewer #2, we now include a paragraph in which we discuss how bikinetic plaid adaptors could be used to address the same question. This is covered in the penultimate paragraph of the Discussion.

Reviewer #2 raised the following minor issues.

1. *Page 10: for the unikinetic condition, why was the moving component drifted at 45 degrees only in the 0-degree case? If I understand this correctly, a unikinetic plaid with a vertical static orientation and an upward drifting component should still be perceived as moving upward. The author would need to justify why they used a different tilt for the static orientation for the 0-degree, unikinetic case, or rerun this case with the above-described setup, so that the 0-degree case in the unikinetic condition could be more fairly compared with the other adaptor directions.*

Response. This configuration permits us to test whether the component motion contributes to the DAE. If we had used a configuration in which the component motion drifted upwards we would not be able to test if it contributed to the DAE, since the test stimulus direction was also upwards. We have re-written the final paragraph in the Introduction to help clarify this point.

2. *What was the speed of the test RDK? This is an important piece of information because DAE is speed-tuned, and this is somewhat related to the Major issue number 1 above.*

Response. The speed of the test RDK was 3° s^{-1} – the same as the adaptor’s pattern speed. This information is now included at end of the third paragraph page 9.

3. *Some details about the procedure need to be added, e.g., exact number of trials in each block, number of blocks completed by each observer, and the order of presentation of the adaptor directions.*

Response. The relevant details have been added to the Procedure section.

4. *There were both clockwise and counter-clockwise (relative to vertical) directions for the non-vertical adaptors, but it was not described how these adaptor directions were presented. I suppose they were alternated or at least randomized across different blocks, but this needs to be clarified.*

Response. Yes, they were randomized across different blocks. This information is now in the Procedure.

5. *I assume the authors took the estimated PSE as DAE magnitude. But this was not explicitly mentioned in method or results. Please add it back.*

Response. DAE magnitude was calculated as the angular difference between the PSE and vertical upwards. This information has been included in the Procedure.

6. *The doi link at the bottom of the manuscript does not work for me.*

Response. The Doi link has been fixed, although it appears that it will only be activated when the manuscript is accepted.

Reviewer #3

Reviewer #3 raised the following issues.

1. *I have some concern that all the participants are authors, so it is possible in principle for bias effects to contribute to the results. The psychophysical method was not bias-free because the response was explicitly*

about the perceived direction of the test pattern. It would be preferable to have data for naïve participants as well.

Response. We have tested three naïve participants. The results remain unchanged.

2. *Results clearly show a major contribution to adaptation from the plaid direction. However, the claim that there is no effect of the local/grating direction is not so convincing. This claim requires that results for the zero-degree adapters in the baseline and plaid conditions should be the same, at zero. On the other hand, the presence of a DAE at a plaid direction of 0 deg would indicate some degree of adaptation to the 45 deg grating direction. The SE bars suggest an above-zero effect in this plaid condition (Fig. 2b). The results report a t-test that there was no significant effect in this condition. It is not clear what this t-test tested. Given that the same participants took part in all sessions, it would be possible to run a t-test to compare the zero deg adapters in the baseline and plaid conditions. Is this the t value reported? Perhaps more (naïve) participants would push this difference into significance?*

Response. The non-significant t-value reported refers to a repeated measures t-test comparing performance following presentation of the zero degree adapters in the baseline and plaid conditions. This is now explicitly stated in the Results section. As recommended by reviewer #3, we tested three additional naïve participants and the result was still not significant.

3. *If adaptation involves multiple components at different levels, one might expect some effect from the grating component. Indeed it could be argued that there is so much evidence for adaptation at multiple levels of adaptation, including V1 and MT, that it is too simplistic to present the predictions in terms of an either-or dichotomy. There is evidence for adaptation in V1, so why would this not contribute to the DAE at all? It could be argued in reply that the grating by itself at 45 deg produced a much larger effect (45 deg in the baseline condition, Fig 2a) than when it was present in the plaid (0 deg in the plaid condition, Fig 2b), but this could reflect the relative strengths of the two adapting components, as well as some inhibitory interaction between the static component and the drifting component. The point is at least worth debating.*

Response. We agreed with Reviewer #3 that, given the evidence for adaptation at multiple levels of adaptation, one might expect adaptation in

V1 to contribute to the observed DAE. We raise this possibility in the final paragraph of the Introduction, and argue in the Discussion (second paragraph of page 16) that there is no evidence that the component motion contributed to the DAE. If it did contribute to the effect, then we would expect DAE magnitude to differ in the baseline and unikinetic plaid conditions. However, as discussed in the Results, there was no significant difference between the baseline and plaid conditions for three of the four directions tested (0° , 22.5° and 66.7°); this suggests that the observed DAE was not a consequence of an interaction between the component and pattern motions, but was driven by the pattern motion.

4. *Have the authors considered the possibility of an adaptation component mediated by second-order motion sensors that encode pattern direction in plaids? A number of papers, and a theoretical model, favour a role for second-order sensors in plaid processing and adaptation effects (e.g. Wilson, H. R., & Kim, J. 1994. A model for motion coherence and transparency. *Visual Neuroscience*, 11(6), 1205-1220; papers by Nishida cited in the manuscript).*

Response. We had not considered this possibility in the current paper, partly because the experiment was not addressing this specific question about the role of second-order sensors in plaid processing. While a number of papers suggest such a role for second-order sensors, our data do not speak directly to this question and we feel that any attempt to do so would be speculative and beyond the remit of the paper.